# Towards the characterization of the hidden world of small proteins in *Staphylococcus aureus*, a proteogenomics approach

Stephan Fuchs[1], Martin Kucklick[2,3], Erik Lehmann[2,3], Alexander Beckmann[2,3], Maya Wilkens[1,2,3], Baban Kolte[4], Ayten Mustafayeva[2,3], Tobias Ludwig[2,3], Maurice Diwo[2,3], Josef Wissing[5], Lothar Jänsch[5], Christian H. Ahrens[6], Zoya Ignatova[4], Susanne Engelmann[2,3]*

1 Robert Koch Institute, Methodenentwicklung und Forschungsinfrastruktur (MF), Berlin, Germany,
2 University of Technical Sciences Braunschweig, Institute for Microbiology, Braunschweig, Germany,
3 Helmholtz Center for Infection Research GmbH, Microbial Proteomics, Braunschweig, Germany,
4 University of Hamburg, Institute of Biochemistry and Molecular Biology, Hamburg, Germany, 5 Helmholtz Center for Infection Research GmbH, Cellular Proteomics, Braunschweig, Germany, 6 Agroscope, Research Group Molecular Diagnostics, Genomics and Bioinformatics & SIB Swiss Institute of Bioinformatics, Basel, Switzerland

* Susanne.Engelmann@helmholtz-hzi.de

**Data Availability Statement:** The mass spectrometry proteomics data have been deposited to the ProteomeXchange Consortium (http://

## Abstract

Small proteins play essential roles in bacterial physiology and virulence, however, automated algorithms for genome annotation are often not yet able to accurately predict the corresponding genes. The accuracy and reliability of genome annotations, particularly for small open reading frames (sORFs), can be significantly improved by integrating protein evidence from experimental approaches. Here we present a highly optimized and flexible bioinformatics workflow for bacterial proteogenomics covering all steps from (i) generation of protein databases, (ii) database searches and (iii) peptide-to-genome mapping to (iv) visualization of results. We used the workflow to identify high quality peptide spectrum matches (PSMs) for small proteins ($\leq$ 100 aa, SP100) in *Staphylococcus aureus* Newman. Protein extracts from *S. aureus* were subjected to different experimental workflows for protein digestion and prefractionation and measured with highly sensitive mass spectrometers. In total, 175 proteins with up to 100 aa (SP100) were identified. Out of these 24 (ranging from 9 to 99 aa) were novel and not contained in the used genome annotation.144 SP100 are highly conserved and were found in at least 50% of the publicly available *S. aureus* genomes, while 127 are additionally conserved in other staphylococci. Almost half of the identified SP100 were basic, suggesting a role in binding to more acidic molecules such as nucleic acids or phospholipids.

## Author summary

Conventional automatic genome annotation algorithms often neglect open reading frames smaller than 300 nucleotides (sORF). There are several reasons hindering

proteomecentral.proteomexchange.org) via the PRIDE partner repository (Vizcaino JA, Csordas A, del-Toro N, Dianes JA, Griss J, Lavidas I, et al. 2016 update of the PRIDE database and its related tools. Nucleic Acids Res. 2016;44: D447-56.) with the dataset identifier PXD017932. High Quality MS/MS spectra of the identified tryptic peptides unique for SP100 are accessible in Supplemental Materials. Ribosome profiling data have been deposited within Gene Expression Omnibus (GEO) under accession number GSE150601.

**Funding:** This work was funded by the Deutsche Forschungsgemeinschaft (https://www.dfg.de/) (GRK PROCOMPAS) to SE, by the Deutsche Forschungsgemeinschaft (GRK PROCOMPAS) to LJ, by the Deutsche Forschungsgemeinschaft (INST 188/365-1 FUGG DFG) to SE, by the Schweizerischer Nationalfonds zur Förderung der Wissenschaftlichen Forschung (http://www.snf.ch/de/Seiten/default.aspx) (197391) to CHA, by the Deutsche Forschungsgemeinschaft (IG 73/16-1 SPP 2002) to ZI. The funders had no role in study design, data collection and analysis, decision to publish, or preparation of the manuscript.

**Competing interests:** The authors have declared that no competing interests exist.

automatic annotation and prediction of short genes: (i) sORFs possess insufficient sequence information for domain and homology search, (ii) only a limited number of experimentally validated sORFs can serve as templates, and (iii) sORFs show the tendency to be species-specific. We thus established a proteogenomics workflow, which is executed by two open source tools, Salt and Pepper (https://gitlab.com/s.fuchs/pepper), and uses peptide data obtained by mass spectrometry for identification of genes in bacteria that are hardly predictable by automatic annotation algorithms. As a proof of concept, we selected *Staphylococcus aureus*, one of the most frequently sequenced bacteria and identified 36 proteins not yet considered in the used genome annotation of *S. aureus* Newman. 24 there of are novel small proteins with up to 100 aa (SP100) in *S. aureus* Newman. This clearly demonstrates that our workflow is ideally suited to improve gene annotation of already annotated bacterial genomes. In the future, it may also facilitate protein and ORF detection in not annotated bacterial genomes.

## Introduction

*Staphylococcus aureus* is a Gram-positive human pathogen of great clinical importance. *S. aureus* causes mainly nosocomial infections in immunocompromized patients, which are frequently associated with difficult to treat multidrug-resistant *S. aureus* phenotypes [1]. With 11,809 genome sequences (including 576 complete genomes), which are publicly available in the reference sequence database of the National Center of Biotechnology Information (RefSeq; status 2020-08-19), *S. aureus* is among the most frequently sequenced bacteria. The number of annotated open reading frames ranges from 2,411 to 3,147 per complete genome sequence. The entire pan-genome of *S. aureus* has not yet been described, due to the fact that the genomic diversity of *S. aureus* is very high [2,3]. However, a preliminary *S. aureus* pan-genome based on the comparison of 64 *S. aureus* genome sequences is composed of 7,411 genes, of which about 20% are conserved constituting the core-genome [3]. The highest variability has been found among genes coding for extracellular and surface-associated proteins [4] which is of particular importance as these proteins are essentially involved in direct interactions with the host environment during infection.

The protein inventory of several *S. aureus* strains has been described using highly sensitive mass spectrometry (MS) techniques combined with liquid chromatography (LC) [5–7]. For *S. aureus* strain COL, more than 1,700 proteins (about 60% of the theoretical proteome) have been identified, quantified and assigned to various subcellular localizations [5,7,8], which can help to predict functions for co-expressed and/or co-localized proteins [9]. However, one group of proteins was highly underrepresented in the *S. aureus* proteome: very small proteins no longer than 100 amino acids (aa) (= SP100). Altogether, Becher and colleagues [5] detected 82 annotated SP100 of which only four proteins were below 50 aa in length (= SP50).

The experimental detection of SP100 by shotgun proteomics is difficult and additionally hampered by the fact that the corresponding short open reading frames (sORFs) are often overlooked by conventional genome annotation algorithms. There are several reasons hindering automated prediction and accurate annotation of sORFs, such as insufficient sequence information for domain and homology searches, a limited number of experimentally validated templates, and their tendency to species specificity [10–12]. Hence, differentiation between sORFs with low and high coding potential is challenging and the number of false positives among predicted sORFs is extremely high [13]. Given these facts, genome annotations

routinely used arbitrary cut-offs for a minimum ORF length of 50 or 100 codons. In addition, the low molecular weight of these proteins complicates experimental isolation and reduces the number of MS-compatible peptides.

Over the last years, various attempts have been made that address one or both of these issues. This includes experimental approaches such as ribosome profiling and proteogenomics to identify this group of proteins as well as bioinformatics approaches for a more reliable prediction and comprehensive annotation of sORFs [14–26]. For instance, different computational approaches have been developed for sORF prediction, which have in common that the coding potential of a putative sORF is scored based on one or more features such as nucleotide composition, synonymous and non-synonymous substitution rates, phylogenetic conservation or protein domain detection [13,16].

Despite these major challenges, there is no doubt that small proteins play a pivotal role in essential cellular processes, hence, it is extremely important to improve our ability to uncover this pool of hidden proteins [20,27–30]. First functional characterizations prove their involvement in various cellular processes such as protein folding, regulation of gene expression, membrane transport, protein modification and signal transduction in different bacteria (for an overview see [30]). In addition, some small proteins have an extracellular function and exhibit toxic or antimicrobial activity. Interestingly, most of the small proteins characterized so far are associated with the cell membrane and are poorly conserved at the sequence level [29]. In *S. aureus*, the most prominent small proteins are phenol soluble modulins with a length of 20 to 40 aa [31] and delta-hemolysin (26 aa) [32]. Phenol soluble modulins possess multiple roles in *S. aureus* pathogenesis by inducing cell lysis of blood cells, stimulating inflammatory responses and influencing biofilm formation (for review see [33]). Delta-hemolysin interacts with membranes of various blood cells, which concentration dependently results in a membrane disturbance and even in cell lysis (for review see [34]). While both phenol soluble modulins and delta-hemolysin have been studied in detail in recent years, data on the identification and functional characterization of other SP50 in *S. aureus* are almost completely missing.

The availability of numerous *S. aureus* genome sequences defines it as a well-suited model bacterium for prediction and identification of small proteins and peptides. The number of annotated coding sequences with up to 303 nucleotides is highly variable in the 576 complete *S. aureus* genome sequences ranging from 287 to 621 SP100. This is mainly attributed to the fact that various algorithms of genome annotation were applied. Among *S. aureus* reference strains, strain Newman plays a pivotal role. First isolated in 1952 from a human infection [35], it is one of the most frequently used *S. aureus* strains in infection models as it is characterized by a relatively stable phenotype. In addition, four prophages were identified in the genome of strain Newman, inserted at different sites in the chromosome, exceeding the regularly observed number of prophages in *S. aureus* [36–38]. In a murine infection model, the loss of all four prophages significantly reduced the virulence potential of the strain [39]. The existence of these prophages made *S. aureus* Newman an excellent model to study their impact on virulence and cells physiology. The genome sequence of strain Newman, was predicted to encode at least 2,854 proteins, again, the number of annotated SP100 is rather low and amounts to 493 proteins [40] (NC_009641.1; genome annotation from 2020-02-17).

To more comprehensively identify proteins with up to 100 aa, we used *S. aureus* Newman as a model system and developed an fully-featured proteogenomics workflow that combines *in silico* translation of the entire genome sequence, various LC-MS/MS workflows, and a bioinformatics pipeline for peptidomics data analyses. The workflow is highly optimized, flexible and ready to use in other bacterial species.

## Methods

### Bacterial strains, cultivation conditions and cell lysis

*S. aureus* Newman [35] was cultivated in 100 mL complex medium (TSB) at 37˚C and 120 rpm to an optical density at 540 nm ($OD_{540}$) of 1 and 7. Cells were harvested by multiple centrifugation steps and disrupted by cell homogenization (FastPrep-24, MP Biomedicals) (for details see [41]). All experiments have been performed with three biological replicates. The protein concentration was determined using the Roti-Nanoquant assay (Roth, Karlsruhe, Germany) and the protein solution was stored at -20˚C.

### Fractionation of proteins and peptides and proteolytic cleavage

**Gel-based approach.**   40 µg of cytoplasmic proteins were separated by one dimensional SDS polyacrylamide gel electrophorese (1D SDS PAGE) according to Laemmli [42] with the following modifications: the loading buffer consisted of 3.75% (v/v) glycerol, 1.25% (v/v) ß-mercaptoethanol, 0.6% (w/v) SDS, 0.0014% (w/v) bromophenol blue, 16.5 mM Tris-HCl (pH 6.8). The separation gel contained 12% (w/v) acrylamide gel (with 0.32% bisacrylamide), 0.375 M Tris-HCL (pH 8.8), 0.255% (w/v) SDS, 0.062% (w/v) APS, and 0.062% (v/v) TEMED and the stacking gel 5% (w/v) acrylamide (with 0.13% (w/v) bisacrylamide), 0.125 M Tris-HCl (pH 6.8) 0.25% (w/v) SDS, 0.075% (w/v) APS, and 0.075% (v/v) TEMED.

Proteins were fixed with 40% (v/v) ethanol and 10% (v/v) acetic acid for one hour and subsequently stained with colloidal coomassie [43] for one hour. In-gel digestion using trypsin and extraction of the peptides were carried out as described by Lerch et al. [43] with an additional extraction step using acetonitrile.

For digestion with Lys-C, a buffer containing 25 mM TRIS/HCl and 1 mM EDTA (pH 8.5) was used. The applied enzyme concentration was 1/40 of the total protein concentration. Digestion of AspN was performed in 10 mM Tris-HCl (pH 8.0) with a final AspN concentration of 1/50 of the total protein concentration.

**Gel-free approach.**   The gel-free approach was performed by applying tryptic in-solution digestion followed by an Oasis HLB-SPE-cartridge purification and SCX-fractionation. In detail, 40 µg of crude protein extract were solved in 8 M urea and 2 M thiourea and adjusted to a final concentration of 6 M urea. After addition of 1.6 µL of 5 mM DTT in 50 mM ammonium bicarbonate solution (pH 7.8), the protein solution was incubated for 30 min at room temperature. For alkylation, 1 µL of a freshly prepared 55 mM IAA in 50 mM ammonium bicarbonate buffer was added to 10 µL of the protein solution and incubated for 20 min in the dark at room temperature. Subsequently, the solution was adjusted to a final concentration of 1 mM $CaCl_2$ and 1 M urea using $CaCl_2$ solved in a 50 mM ammonium bicarbonate buffer. For digestion, 1 µg trypsin (in 50 mM ammonium bicarbonate and 1 mM $CaCl_2$) was applied for 50 µg protein. Digestion was performed for 12 h at 37˚C with gentle agitation (50 rpm) and stopped by acidification to a pH value of $\leq$ 2.5 with 10% formic acid.

For peptide purification, Oasis HLB-SPE-cartridges (1cc, 10mg, Waters, Milford, MA, USA) were initially conditioned with acetonitrile and then with 0.5% formic acid in 60% acetonitrile. Subsequently, cartridges were equilibrated with two volumes of 0.5% formic acid. Samples were loaded on the cartridges; the flow-through was collected and again loaded on the cartridge. The peptides were washed five times with 0.5% formic acid (FA) and eluted twice with 0.85 mL 60% ACN 0.5% FA. Eluates were dried in a speedvac (Eppendorf Concentrator plus, Eppendorf AG, Hamburg, Germany) and frozen at -20˚C.

SCX fractionation was done as described by Kummer et al. [44]. To reduce the number of fractions to eight, peptide-containing fractions were combined.

**Peptide desalting.** ZipTips (C18, Merck Millipore, Billerica, MA, USA) were conditioned with 50% acetonitrile twice. Subsequently they were equilibrated three times with 0.1% FA in 5% acetonitrile. 10 μL of each peptide fraction (resolved in 20 μL 0.1% FA in 5% acetonitrile for 60 min) were loaded on the C18 matrix of the tip by aspirating 10 times. Elution was performed three times by aspirating five times with 0.1% FA in 60% acetonitrile in a new micro test tube. Samples were dried in a speedvac.

## Liquid chromatography coupled mass spectrometry (LC-MS/MS)

For LC-MS/MS, each peptide fraction of a sample was solved in 16 μL of 0.1% FA in 3% acetonitrile for one hour, ultrasonicated in a water bath for 5 min and ultracentrifuged.

**Orbitrap Velos Pro MS.** LC-MS/MS runs with the Orbitrap Velos Pro MS (Thermo Fisher Scientific Inc, Waltham, MA USA) were done as described by Lerch and coworkers [43].

**Orbitrap Fusion MS.** LC-MS system and used columns are described by Bulitta and coworkers [45]. A 200 min gradient was applied, starting with 3.7% buffer B (80% acetonitrile, 5% DMSO and 0.1% formic acid) and 96.3% buffer A (0.1% formic acid, 5% DMSO): 0–5 min 3.7% B; 5–125 min 3.7–31.3% B; 125–165 min 31.3–62.5% B; 165–172 min 62.5–90.0% B; 172–177 min 90% B; 177–182 min 90–3.7% B, 182–200 min 3.7% B.

Primary Scans were performed at the Orbitrap in the profile modus scanning an m/z of 350–1800 with a resolution (full width at half maximum at m/z 400) of 120,000 and a lock mass of 445.1200. Using the Xcalibur software (Thermo Fisher Scientific Inc., San Jose, CA, USA), the mass spectrometer was controlled and operated in the "top speed" mode, allowing the automatic selection of as much as possible twice to fourfold-charged peptides in a three-second time window, and the subsequent fragmentation of these peptides. In the non-targeted modus, primary ions (±10 ppm) were selected by the quadrupole (isolation window: 1.6 m/z), fragmented in the ion trap using a data dependent CID mode (top speed mode, 3 seconds) for the most abundant precursor ions with an exclusion time of 13 s and analysed by the ion trap.

## Protein database generation

To consider its full coding potential, the genome sequence of the *S. aureus* subsp. *aureus* strain Newman (NC_009641.1) was translated in all six reading frames from stop to stop codon using the SALT tool (https://gitlab.com/s.fuchs/pepper). Genome circularity has been considered and entries with less than 9 amino acids excluded resulting in a total of 177,532 sequence entries.

## MS Data analysis and statistics

Analyses of the obtained MS and MS/MS data were performed using MaxQuant (Max Planck Institute of Biochemistry, Martinsried, Germany, www.maxquant.org, version 1.5.2.8) and the following parameters: peptide tolerance 5 ppm; a tolerance for fragment ions of 0.6 Da; variable modifications: methionine oxidation and acetylation at protein N-terminus, fixed modification: carbamidomethylation (Cys); a maximum of two missed cleavages and four modifications per peptide was allowed. For the identification of SP100, a minimum of one unique peptide per protein and a fixed false discovery rate (FDR) of 0.0001 for PSMs and 0.01 for proteins was applied. The minimum score was set to 40 for unmodified and modified peptides, the minimum delta score was set to 6 for unmodified peptides and to 17 for modified peptides. All samples were searched against the *S. aureus* Translation Database (TRDB) with a decoy mode of reverted sequences and common contaminants supplied by MaxQuant

For identification of non-annotated open reading frames based on identified peptides a proteogenomics tool has been developed that describes any identified peptide in the degenerated DNA code sequence. By this, exact matches within the reference genome can be found and, subsequently, filtered based on existing annotation and location.

### Phylogenetic and functional analyses

For phylogenetic analyses we downloaded all complete genome sequences for *S. aureus* (n = 541) and staphylococci (n = 165) from NCBI RefSeq (state of 2020-05-18). All SP100 sequences were searched against the downloaded genome sequences using tblastn. Based on the best hit alignment to every bacterial chromosome, the identity related to the full query length was calculated. Only alignments sharing at least 90% identity with the full query sequence were considered. Based on this, relative species and genus conservation rates have been calculated.

For functional analyses we searched all SP100 sequences against the eggNOG database v5.0 using eggNOG-mapper 2.0 (default parameters). The taxonomic scope was automatically adjusted to each query to ensure correct classification of phage proteins. Only functions from one-to-one orthology were transferred.

### Ribosome profiling

**Library preparation.**  *S. aureus* cells from 30 mL culture grown in TSB medium to $OD_{550}$ = 1 were harvested by rapid centrifugation and resuspended in 390 μL ice cold 20 mM Tris lysis buffer pH 8.8, containing 10 mM $MgCl_2$ x 6 $H_2O$, 100 mM $NH_4Cl$, 20 mM Tris (pH 8.0), 0.4% Triton-X-100, 4 U DNase, 0.4 μL Superase-In (Ambion), 1mM chloramphenicol. Cells were disrupted by cell homogenization (FastPrep-24, MP Biomedicals) with 0.5 mL glass beads (diameter 0.1 mm) for 30 s at 6.5 m/s followed by incubation on ice for 5 min. These steps were repeated twice. To remove cell debris, cell lysates were centrifuged and subsequently stored at -80˚C and 100 $A_{260}$ units of ribosome-bound mRNA fraction were subjected to nucleolytic digestion with 10 units/μl micrococcal nuclease (Thermofisher) in buffer with pH 9.2 (10 mM Tris pH 11 containing 50 mM $NH_4Cl$, 10 mM $MgCl_2$, 0.2% triton X-100, 100 μg/mL chloramphenicol and 20 mM $CaCl_2$). The rRNA fragments were depleted using *S. aureus* riboPOOL rRNA oligo set (siTOOLs, Germany) and the library preparation was performed as previously described [46].

**Bioinformatic analyses of ribosome profiling RNAs.**  Raw sequencing reads were trimmed using FASTX Toolkit (quality threshold: 20) and adapters were cut using cutadapt (minimal overlap of 1 nt) and mapped to the genome version NC_009641.1 (NCBI, January 2020). Following extraction of reads mapping to rRNAs, the remaining reads were uniquely mapped to the reference genome using Bowtie, parameter settings: -l 16 -n 1 -e 50 -m 1—strata–best y. Non-uniquely mapped reads were non-considered (for more details see [47]).

## Results

### Estimating the number of spurious sORFs in *S. aureus* Newman using single-nucleotide permutation testing

A single nucleotide permutation test was used to verify the global confidence and significance of ORFs based on their length only. For this purpose, ORFs were detected in the genome sequence of *S. aureus* Newman (NC_009641.1; NCBI translation table 11; longest ORFs preferred). As false-positive estimate, we used the median number of ORFs detected in permuted genome sequences (n = 1000), that show the same nucleotide composition but in random

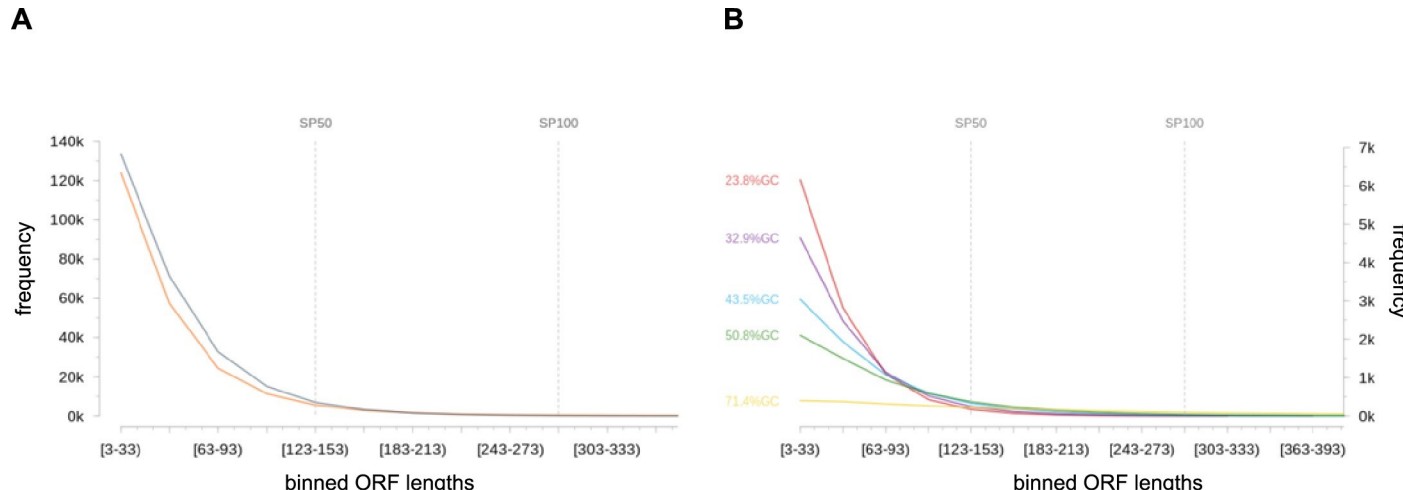

**Fig 1. sORF frequencies in biological and permuted genome sequences. (A) Estimation of the proportion of false-positive sORFs in predictions based solely on start and stop codons:** All potential ORF sequences (NCBI translation table 11; longest ORF variants preferred) were extracted from the genome sequence of *S. aureus* Newman and 1,000 permuted sequence derivatives showing the same nucleotide composition but in random order and can therefore be assumed to no longer contain any biological information. Resulting ORFs were binned based on their length. Bin sizes are shown for the genuine reference sequence (orange) and the permuted sequences (grey; as median) up to a maximum ORF length of 303 bp (= 100 aa). Especially small ORFs tend to occur randomly. **(B) Impact of GC content on the number of spurious ORF:** According to (A), bin sizes are given for genome sequences and their permuted sequence derivatives (n = 1,000; as median) with varying GC content. The used reference genome sequences were NC_000913.3 (*Escherichia coli* K-12 substr. MG1655; 50.8%GC; 4,641,652 bp), NC_000964.3 (*Bacillus subtilis subsp. subtilis* str. 168; 43.5%GC; 4,215,606 bp), NC_007633.1 (*Mycoplasma capricolum subsp. capricolum* ATCC 27343; 23.8%GC, 1,010,023 bp), NC_009641.1 (*Staphylococcus aureus subsp. aureus* str. Newman; 32.9%GC; 2,878,897 bp), NC_010162.1 (*Sorangium cellulosum* So ce56; 71.4%GC; 1,303,779 bp).

order and can therefore be assumed to not contain any biological information (Fig 1A). Accordingly, the false discovery rate (FDR) of ORFs detected in the biological sequence is 113% for those with a maximum length of 63 bp (coding for 20 aa), 133% for those with a length between 66 and 153 bp (21 to 50 aa) and 94% for those with a length between156 and 303 bp (51 to 100 aa). This highlights the need for additional evidence for the reliable annotation of sORFs. In contrast, coding sequences (CDS) with a length of at least 396 bp ($\geq$ 131 aa) did not occur by chance in the permuted sequence set. Since start and stop codons tend to be AT-rich, the FDR for sORFs increases with an increasing GC content of an organism (Fig 1B).

### Creating more comprehensive protein databases for *S. aureus* Newman using *in silico* translation

To identify small proteins not covered in the RefSeq annotation, we generated a protein database considering the full coding potential of *S. aureus* Newman by translating all six reading frames of the respective genome sequence and creating a separate protein entry for each sequence between two stop codons. The resulting **TR**anslation **D**ata**B**ase (TRDB) comprises 177,532 sequences with a minimum length of 9 aa. We established an automated workflow for the translation of (circular) bacterial genome sequences (Fig 2A). The corresponding python-based tool called Salt is publicly available (https://gitlab.com/s.fuchs/pepper). It allows the extraction of all potential ORFs from a circular or linear genome sequence and supports both start to stop and stop to stop codon extraction (according to NCBI translation table 11). The extracted sequences can be automatically translated and, if required, digested *in silico* into individual peptides using predefined digestion patterns of different enzymes. All DNA, protein and peptide sequences can be stored in individual FASTA files. The respective sequence header can be fully customized to meet specific requirements. Moreover, for each sequence collection, feature tables can be exported as tabulator delimited text files listing different physicochemical

## A. Salt

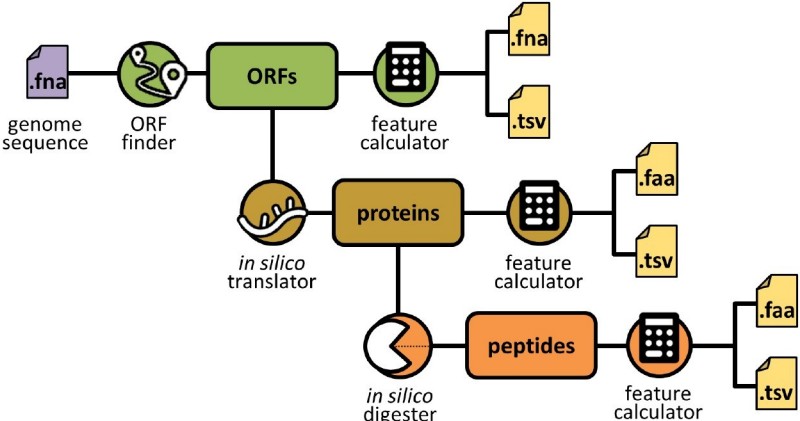

## B. Pepper

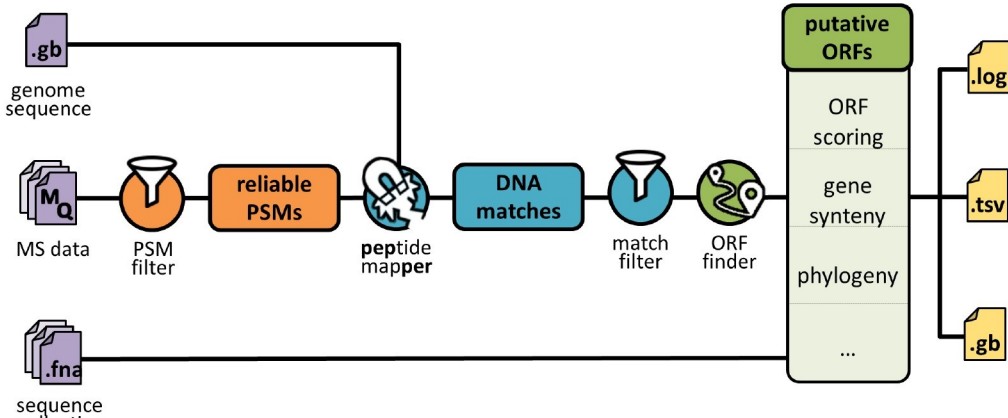

## C. Result Visualization

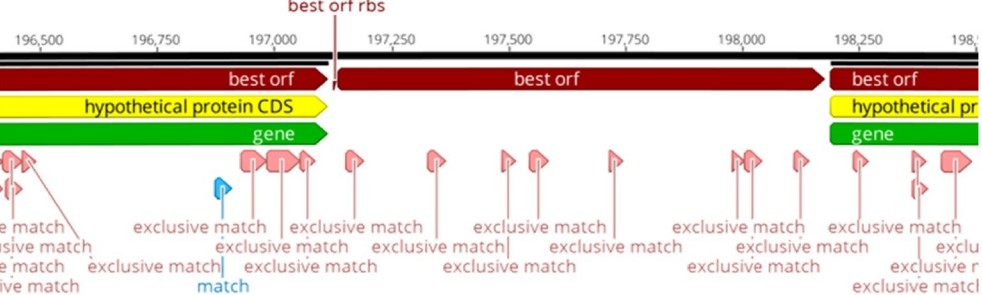

**Fig 2. Bacterial proteogenomics workflow provided by Salt and Pepper. (A) Creation of protein and peptide databases using Salt:** Based on a FASTA file as input, Salt extracts all potential ORFs from a given (circular) genome sequence using different methods (stop to stop codon, start to stop codon). The resulting ORF sequences are then translated *in silico* into protein sequences that can be further digested using different *in silico* proteases (Tryspin, Chymotrypsin, Asp-N, Lys-C and Proteinase K). For each level (ORFs, proteins, peptides) individual FASTA files and tables (tab-separated values, TSV) are created, listing various sequence-derived properties such as molecular weight or isoelectric points. **(B) Proteogenomics analyses using Pepper:** Peptide to spectrum matches (PSMs) obtained from different samples are directly extracted from MaxQuant evidence files (MQ). Spectral data can be extensively evaluated

to apply different spectrum quality and replication criteria can be defined to restrict the analysis to highly-reliable PSMs only. Respective coding sites are determined in a given (circular) genome sequence provided as FASTA file. The resulting coding sites (DNA matches), that can be filtered e.g. by exclusivity are used to predict the putative open reading frames. Additional information such as potential ribosomal binding sites, gene synteny based on the reference genome annotation, and conservation in given sequence collections (provided as FASTA files) are collected. Different files are created to archive all analysis parameters (log file), results on peptide, DNA match, and ORF level (TSV), and an updated reference genome annotation integrating the identified DNA matches and ORFs (Genbank file; GB). **(C) Results visualization:** GB files created by Pepper can be used for results visualization using third-party software (here: Geneious Prime, Biomatters Ltd.). The genome sequence (black line) with coordinates is shown on top. Existing annotations are highlighted in yellow and green. ORFs with the highest coding potential regarding Pepper and potential ribosomal binding sites (RBS) are highlighted in red. DNA matches are show in light-red, if the respective peptide is not encoded elsewhere in the genome (exclusive match), or light-blue, if multiple coding sites exist for the respective peptide.

properties such as molecular weights, isoelectric points or grand average of hydropathy (GRAVY) values. To the best of our knowledge, this is the only freely available tool that offers such a variety of functions (further information see https://gitlab.com/s.fuchs/pepper).

## Creating a fully automated yet flexible workflow for bacterial proteogenomics

To deduce putative ORFs from a list of identified peptides, we created a rule-based expert system called Pepper which is fully automated and optimized for bacterial proteogenomics (Fig 2B). In brief, sequences and quality measures of peptide spectrum matches (PSMs) are extracted from evidence files (and, optionally, ms/ms files) provided by the MaxQuant software (Max Planck Institute of Biochemistry, Martinsried, Germany, version 1.5.2.8; http://www.maxquant.org). Different spectrum- and quality-based filter criteria can be then applied automatically to restrict the analyses to high-quality PSMs only (at least 5 consecutive y- or b-ions **or** at least 2 * 4 y-ions or b-ions **or** at least 4 b- and 4 y-ions) (see also Table 1). The respective filter criteria were deduced from a very extensive visual inspection and assessment of the MS/MS spectra by experts. The objective was to reduce the number of false-positives when applying a cut-off of only one unique peptide per protein for identification of SP100. We required that there be a sequence tag of at least five consecutive b or y fragment ions or two times four consecutive b or y ions in a spectrum to be considered [21]. In addition, peptide specific mass tracks should have significant levels above the background levels. Hence, Pepper is able to automatically score and filter high-quality PSMs on these specific requirements and to apply additional filters related to the intensity coverage ($> 0.1$), Andromeda Score ($\geq 40$) and posterior error probability ($< 0.1$) (Table 1). High-quality MS/MS spectra for tryptic peptides unique for SP100 in *S. aureus* are accessible in the Supplemental Materials.

In the next step, all potential coding sites (DNA matches) are identified for each high-quality PSM within the given genome sequence considering the degenerated nature of the DNA code. On the basis of the DNA matches found, potential ORFs are deduced according to the following rules: first, an ORF must contain all successive DNA matches that are encoded on the same strand and in the same reading frame and not separated by an interposed stop codon. Secondly, the ORF is extended until the first stop codon downstream of the last DNA match and encoded in the same reading frame. In the final step, predicting the translational start site, the most upstream DNA match covered by the potential ORF plays an essential role. Three different cases can be distinguished here: (1) The identified peptide encoded by the most upstream DNA match is not a proteolytic (e.g. tryptic) product. In this case, the first codon of the most upstream DNA match is assumed to be the translation start site. Since N-terminal methionine residues are cleaved from a number of bacterial proteins during

**Table 1. Filter and ranking options provided by Pepper.**

| option | description | this study[1] |
|---|---|---|
| **PSM filter** | | |
| –minsample $<n>$ | only PSMs identified in $<n>$ or more samples or replicates will be considered | 2 |
| –perrp $<n>$ | only PSMs with a posterior error smaller than $<n>$ are considered | 0.1 |
| –score $<n>$ | only PSMs with a Andromeda score smaller than $<n>$ are considered | 40 |
| –no_msms | PSMs without MS/MS data are not excluded | - |
| –msms $<f>$ | only high-quality PSMs are considered based on MS/MS ions provided in file $<f>$[2] | + |
| –minintcov $<n>$ | only PSMs with a MS/MS intensity coverage of at least $<n>$ are considered | 0.1 |
| –seqfilter $<f>$ | only PSMs referring to a sequence listed in file $<f>$ are considered | - |
| **Coding site filter** | | |
| –uniq_only | multiple coding sites for the same PSM are excluded | - |
| –uniq_only_smart | multiple coding sites for the same PSM are excluded if they do not refer to a gene family[3] | - |

[1]filter $<value>$ used in this study; + indicates that the respective option was used without any value; - indicates that the respective option was not used

[2]high-quality PSMs need to be supported by at least one MS/MS spectra meeting one of the following criteria

- at least 5 consecutive y- or b-ions
- at least 2 * 4 y-ions
- at least 2 * 4 b-ions
- at least 4 b- and 4 y-ions

[3]a gene family is defined by 100% sequence identity including sequences of different lengths (gene truncations)

maturation, the translational start is shifted 3 bp upstream if directly preceded by a methionine encoding codon. (2) If the identified peptide encoded by the most upstream DNA match is of proteolytic origin, the assumed ORF is extended to the first upstream codon that is encoded in the same reading frame and directly preceded by a stop codon (primary start). Then ORF variants are built starting from each (alternative) start codon between primary start and the first (most upstream) DNA match. The different ORF variants are evaluated based on different measures and features such as presence and location of a ribosomal binding site or product length (S1 Table). Comprehensive information collected on spectrum, peptide, and DNA level (S2 Table) is stored in different file formats such as CSV and Genbank. The latter can be used for intuitive result visualization using third party software (Fig 2C). Pepper is open-source and freely available under https://gitlab.com/s.fuchs/pepper.

## Small protein identification using different MS-based approaches

To identify small proteins in *S. aureus*, cytoplasmic proteins prepared from cells grown in complex medium to an optical density ($OD_{540}$) of 1 and 7 (both in three biological replicates) were each subjected to different experimental setups for LC-MS-based protein identification. First, two different techniques for pre-fractionation of proteins/peptides were applied: a gel-based and gel-free approach. For the gel-based approach, proteins were separated by one-dimensional SDS polyacrylamide gel electrophorese (1D SDS PAGE) followed by tryptic in-gel digestion (Fig 3). The gel-free approach was based on tryptic in-solution digestion of proteins followed by an Oasis HLB solid phase extraction-cartridge purification and strong cationic exchange (SCX)-fractionation. In both methods, the resulting peptide fractions were subsequently analyzed by a nanoAQUITY UPLC System coupled to an LTQ Orbitrap Velos Pro

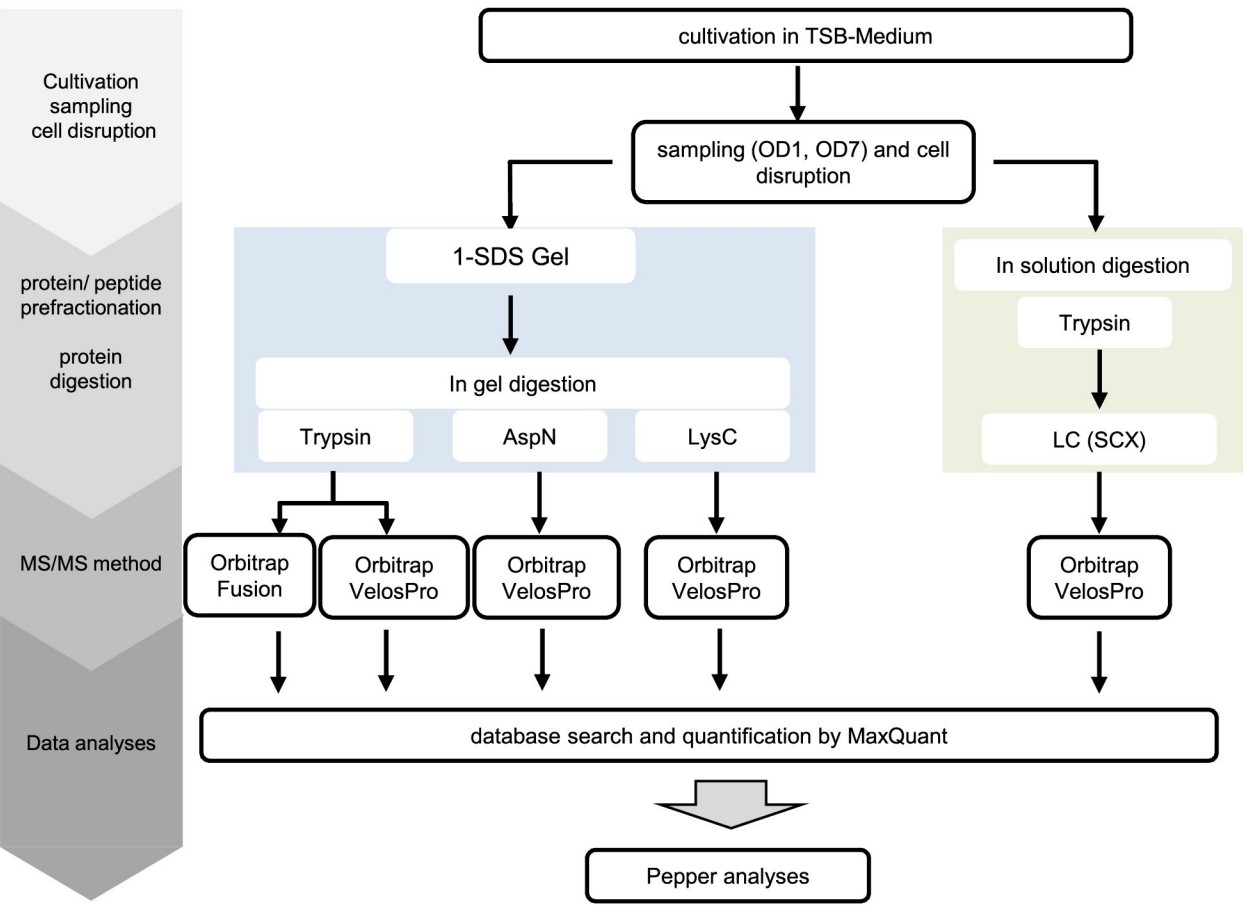

**Fig 3. Mass spectrometry (MS) based identification of small proteins in *S. aureus* Newman.** The different experimental strategies that have been applied for identification of small proteins (SP100) in *S. aureus* Newman are depicted. Cytoplasmic protein extracts of *S. aureus* Newman grown in TSB medium to an optical density of 1 and 7 were prepared from three biological replicates and aliquots of each replicate were used for the different experimental workflows.

mass spectrometer. In addition, we tested the applicability of a second LC-MS system for the identification of SP100: the Orbitrap Fusion MS coupled to a Dionex Ultimate 3000 UPLC System (Thermo Fisher Scientific Inc., Waltham, Massachusetts, USA). For that purpose, the cytosolic protein extracts were pretreated using the gel-based approach (Fig 3).

MS- and MS/MS-data of all samples were searched against the newly established *S. aureus* database TRDB using MaxQuant as an analysis program (Fig 3). Due to their small size the probability to detect more than one unique peptide for proteins with up to 100 aa is relatively low (see also [21,48]. To rely on high-quality peptides only, we modified our MS/MS parameters used by MaxQuant as follows: minimum score > 40 for unmodified and modified peptides, delta score > 6 for unmodified peptides and >17 for modified peptides, peptide spectrum matching (PSM) FDR 0.0001. Combining results of all analyses, 28,153 unique high-quality peptides have been identified, which were further analyzed using our proteogenomics workflow Pepper. To reduce the number of false positives, an additional automated filtering of the MS/MS spectra based on the number of consecutive fragment ions, the intensity coverage, the Andromeda Score and the posterior error probability was applied by Pepper (for details see above and Table 1), which eliminated almost one third of the peptides (n = 8.867). The remaining peptides (n = 19,286) that fulfilled the stringent MS/MS quality criteria (Table 1)

were subsequently mapped to the genome sequence of *S. aureus* Newman by identifying potential coding sites. Peptides, which mapped more than once and were not assigned to protein families by MaxQuant or to the same protein, were manually removed from the list. This resulted in 19,270 unique peptides. Among them 19,195 peptides were allocated to protein sequences derived from 1,901 of 2,854 already annotated open reading frames of *S. aureus* Newman (NC_009641.1; genome annotation from 2020-02-17). Another nine peptides supported an extended 5′-end of the respective ORF. The remaining 66 peptides matched to protein sequences that had not been considered in the annotation of the *S. aureus* Newman genome sequence to date (NC_009641.1; genome annotation from 2020-02-17).

To focus on not yet annotated proteins, we relied on ORFs with the highest coding potential based on different features and quality measures (Tables 1 and S1). In this way, 28 ORFs have been newly described for *S. aureus* Newman ranging between 9 and 741 aa using the different MS-based approaches with tryptic peptides. Regarding SP100, these approaches resulted in 165 SP100 of which 19 were novel.

## Comparing the applied MS based-approaches for small protein identification

To identify which of the applied methods and databases is most powerful for identification of SP100, the obtained identification rates were evaluated. Comparing gel-based and gel-free prefractionation methods showed that both methods support the identification of SP100 (142 versus 124 proteins). The majority of SP100 (n = 109) has been recovered with both pre-fractionation methods, while only a small number of proteins were detected by one method only (Fig 4A and 4B and S3 Table). The usage of the Orbitrap Fusion MS in combination with the gel-based approach for protein pre-fractionation led to the identification of 141 SP100 compared to the 124 proteins using the Orbitrap Velos Pro MS combined with the same protein and peptide fractionation method. In total, 114 SP100 were identified with both instrument combinations (Fig 4A and 4B and S3 Table). These results suggest that the here applied protein/peptide

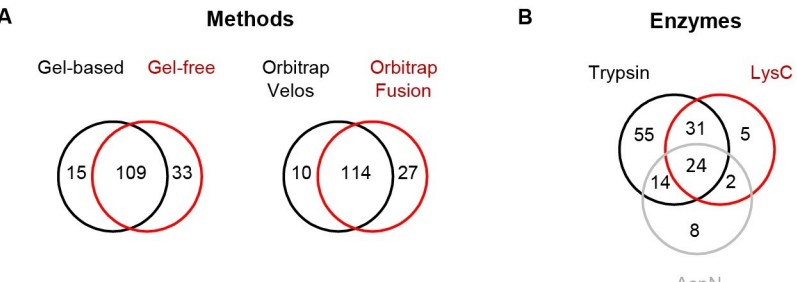

**Fig 4. Comparison of the applied experimental strategies for small protein identification.** Overlaps of identified SP100 are shown. **(A)** The impact of different prefractionation protocols (gelbased and gelfree), LC-MS instruments (Orbitrap Velos and Orbitrap Fusion MS) was analyzed. To test the impact of protein/peptide prefractionation methods, protein extracts were separated on a one-dimensional SDS gels followed by tryptic in-gel digestion of the proteins within the gel fragments for the gel-based approach. For the gel-free approach tryptic in-solution digestion of proteins followed by an Oasis HLB-SPE-cartridge purification and SCX-fractionation has been performed. The resulting peptide fractions were subsequently analyzed by a nanoAQUITY UPLC System coupled to an LTQ Orbitrap Velos Pro MS. To evaluate the impact of different LC-MS systems, an Orbitrap Fusion MS coupled to a Dionex Ultimate 3000 UPLC System and an LTQ Orbitrap Velos Pro coupled to a nanoAQUITY UPLC System have been used to analyse protein extracts pretreated by the gelbased approach. For MS data analyses, the *S. aureus* Newman Translational Database (TRDB) has been used for each experimental strategy. **(B)** To study the impact of different proteases on small protein identification, protein extracts were pretreated by the gelbased approach. In gel digestion has been performed using trypsin, AspN and LysC and the resulting peptides were analysed by a nanoAQUITY UPLC System coupled to an LTQ Orbitrap Velos Pro MS. For peptide identification TRDB has been used.

prefractionation methods and MS instruments were equally well suited for identification of small proteins in *S. aureus*. However, the comparison also revealed that multiple approaches have to be applied to identify the entire set of SP100 in a given protein sample using trypsin for peptide generation. This is in line with other studies showing that application of different approaches enhances the number of identified SP100 [15,19,49]

## Impact of different proteases on small protein identification

For characterization of the identified SP100, Pepper provides several characteristics for each identified protein such as theoretic isoelectric point (pI) and hydrophobicity values (GRAVY) (Fig 5A–5D). Our data revealed that nearly one third of the SP100 identified by our MS based approach depict basic proteins with a pI ≥ 9 (Fig 5B). This percentage is significantly higher compared to proteins larger than 100 aa where only 17.5% showed these characteristics and is even more impressive when only considering proteins with up to 50 aa (SP50). Here, for 60% of the identified proteins a pI ≥ 9 was determined. Digestion of proteins by trypsin results in peptides that end with a basic lysine or arginine. In order to exclude any bias, we studied the influence of different endoproteases on the number and pI of the identified SP100. We used the gel-based approach for protein pre-fractionation and the LTQ Orbitrap Velos Pro MS-based system. Besides trypsin, we selected two additional proteases, Lys-C and AspN, for in-gel protein digestion (Fig 3). MS/MS data analyses have been performed as described above. Using described peptide identification criteria, trypsin digestion resulted in the largest number of peptide identifications (n = 10,477), followed by AspN (n = 5,641) and Lys-C (n = 3,003). The percentage of peptides with a pI ≥ 8 was between 6% (for AspN) and 16% (for Lys-C) (Table 2). Next, we examined the number of identified proteins with specific focus on basic proteins (pI ≥ 8). Accordingly, trypsin resulted in 1,677 protein identifications of which 23% were basic (pI ≥ 8). For Lys-C and AspN, we identified 1,165 and 1,264 proteins and the percentage of identified basic proteins was 21% and 14% (Table 2). Accordingly, the impact generated by the use of different endoproteases on the overall identification of alkaline proteins was marginal. However, considering only identified SP100 and relevant peptides, trypsin and Lys-C clearly supported identification of alkaline proteins. With 41% and 39% the percentage is twice as high as that for all identified proteins. Notably, this is not true for the peptides used for the identification of these proteins. Here the proportion is similar to that found for all identified proteins suggesting that some of the very alkaline SP100 have been identified by acidic or neutral peptides. For *S. aureus* Newman, AspN is the least suited enzyme to identify SP100 under the tested growth conditions, although twice as many peptides have been detected compared to Lys-C. This correlates with a significantly lower percentage of alkaline peptides used for SP100 identification (2%) (Table 2) and may support our hypothesis that SP100 tend to be basic. Using Lys-C and AspN, we additionally expected to increase the average length of the identified peptides and thereby enhancing the sequence coverage and the significance of identification results. However, the average length of the identified unique peptides following the digestion with the selected endoproteases was similar, which is in line with observations made by others testing different proteases for proteomics (Table 2) [50].

In total, we identified 139 SP100 by using different proteases. As expected, the highest overlap was found for trypsin and Lys-C (n = 55 proteins) followed by AspN and trypsin (n = 38 proteins).

## Sequence and expression-based characterization of identified SP100

By applying the different MS-based approaches and the three endoproteases in combination with our optimized proteogenomics workflow we were able to identify 175 SP100 including 14

## A. exclusive peptides

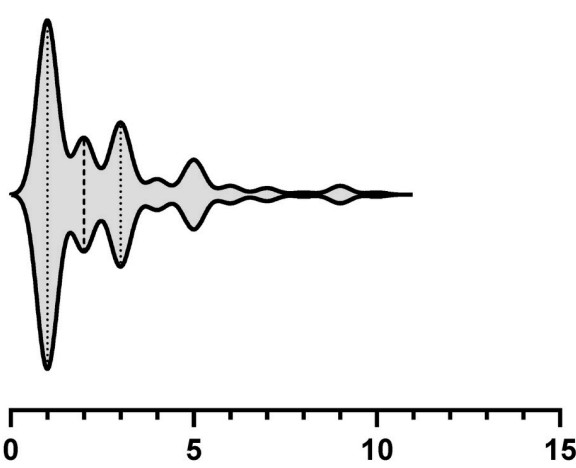

## B. theoretic isoelectric point

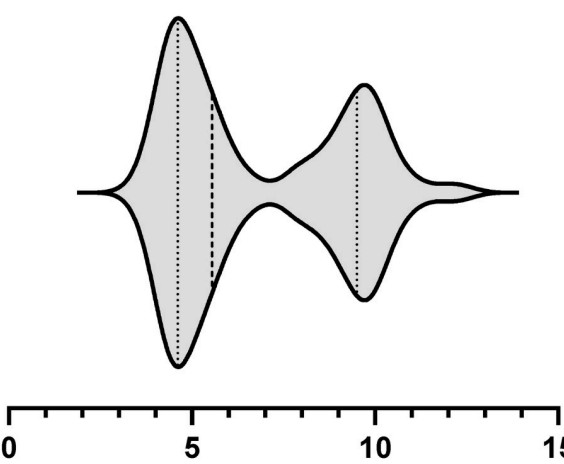

## C. theoretic molecular weight [kDa]

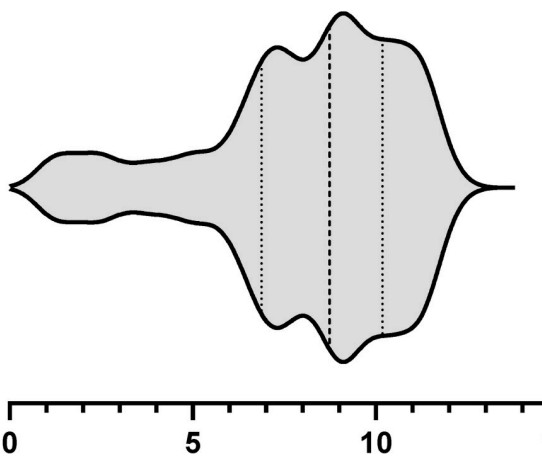

## C. theoretic GRAVY

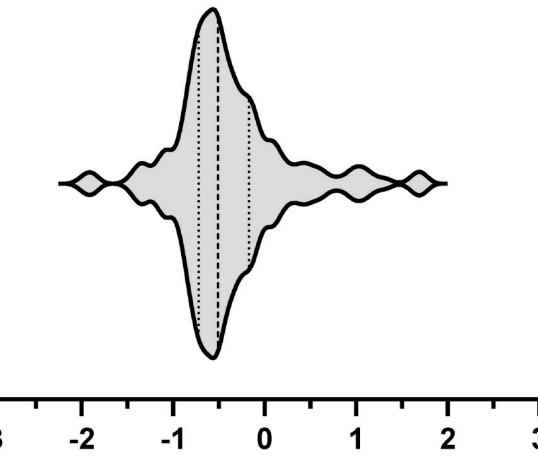

**Fig 5. Characteristics of the identified SP100.** 175 SP100 have been identified in *S. aureus* Newman using the different approaches. For each protein, the number of identified unique peptides (**A**), the isolelectric point (**B**), the molecular weight (**C**) and the hydrophobicity index Gravy (**D**) was determined. The number of proteins with the respective characteristic shown on the x axis are depicted in violin diagrams. Median (dashed line), 50% quantile (dotted line).

very small proteins with up to 25 aa (Fig 6). This study is restricted to soluble intracellular SP100 expressed during growth in complex medium and therefore only provides a snapshot of all SP100 encoded by the genome of *S. aureus* Newman. Many more SP100 are expected among the other protein fractions (e.g. membrane and extracellular proteins) as shown for other bacteria [12,16,17,20,29,51–53]. On the way towards the entire world of SP100 in *S. aureus* it is worth investigating these sub-proteomes using techniques optimized for the identification of small proteins and the here presented proteogenomics workflow.

**Table 2. Identified proteins and peptides using different endoproteases.**

| protease | unique peptides | | | identifed proteins | | unique peptides SP100 | | identified SP100 | |
|---|---|---|---|---|---|---|---|---|---|
| | absolute number | average length (aa) | alkaline[1] (%) | absolute number | alkaline[1] (%) | absolute number | alkaline[1] (%) | absolute number | alkaline[1] (%) |
| Trypsin | 10477 | 13 | 14 | 1677 | 23 | 203 | 16 | 104 | 41 |
| LysC | 3003 | 12 | 16 | 1165 | 21 | 98 | 12 | 69 | 39 |
| AspN | 5641 | 13 | 6 | 1264 | 14 | 82 | 2 | 48 | 19 |

[1] pI ≥ 8

Conservation of the identified sORFs among different *S. aureus* strains as well as among staphylococci was investigated. Accordingly, 144 of these proteins are conserved to at least 90% in half of the sequenced *S. aureus* isolates analyzed (n = 541) and 127 are additionally conserved in other staphylococci (S4 Table and Fig 7). Among the newly discovered proteins 10 are conserved in 50% of the sequenced *S. aureus* isolates and five were also encoded in other staphylococci providing additional support for their existence. Using the eggNOG database, 60 proteins (34%) were successfully assigned to a functional cluster. This revealed an enrichment in functions related to translation, transcription, energy production and replication. However, for 78 SP100 no functional category was assigned (Fig 8). As already mentioned, we observed a significant proportion of alkaline SP100 suggesting that SP100 tend to interact with acidic molecules such as lipids and nucleotides. Moreover, 10 of the identified SP100 were predicted to be integrated into the membrane using LocateP [54], which was supported by the existence of at least one membrane spanning domain (http://www.cbs.dtu.dk/services/TMHMM/).

24 out of the 175 SP100 were novel and not contained in the used genome annotation of *S. aureus* Newman including 16 SP50. The corresponding sORFs were predicted on the basis of the identified high-quality PSMs using the prediction algorithm applied by Pepper. For 10 novel SP100, N-terminal peptides were detected. Only six of these peptides started with

## protein length [aa]

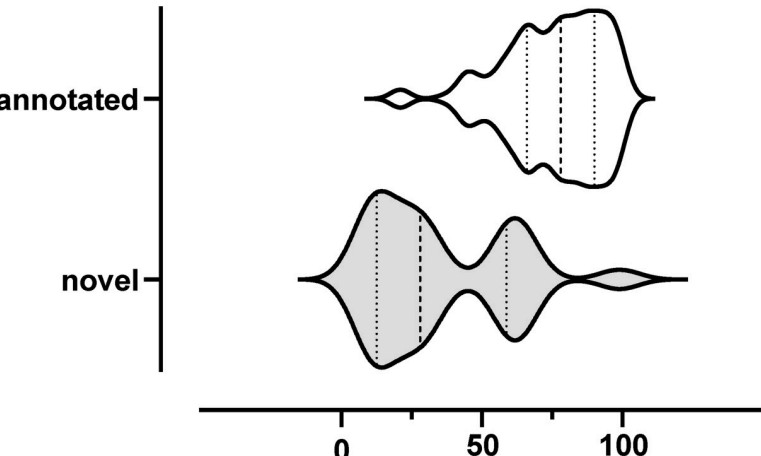

**Fig 6. Lengths of identified SP100.** The violin plot shows the length distribution of annotated and novel SP100 experimentally validated in this study. The respective median is highlighted as dashed line and 50% quantiles as dotted lines.

# phylogenetic conservation [%]

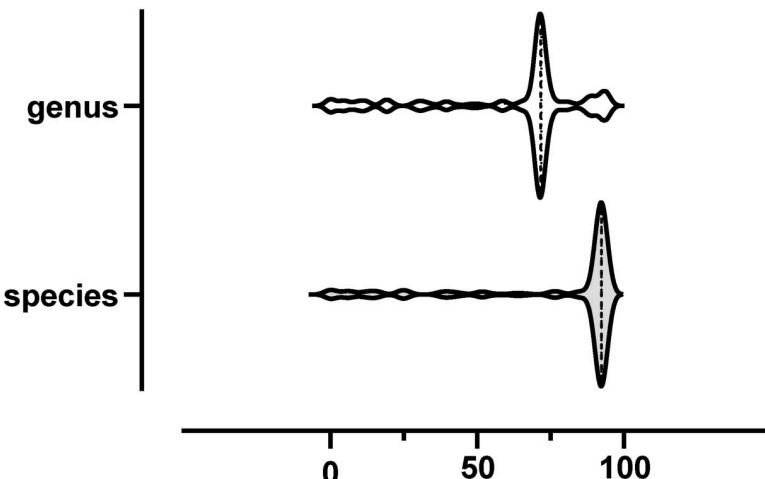

**Fig 7. Phylogenetic conservation of the identified SP100 at species and genus level.** SP100 were searched against the RefSeq genome sequences using tblastn. Based on the best hit alignment to every genome, the identity related to the full query length was calculated. Only alignments sharing 90% identity with the full-length query sequence were considered. On the basis of these results, relative species and genus conservation rates have been calculated.

methionine. For the remaining 14 sORFs, any possible in-frame start codon was assigned and the resulting ORFs were evaluated based on different criteria implement by Pepper (S1 Table). While the majority of the 24 new sORFs has been suggested to start with ATG (n = 11), six sORFs presumably initiate at non-canonical translational start sites such as TTG (n = 2), ATT (n = 1), GTG (n = 1), ATA (n = 1) and ATC (n = 1). For seven sORFs, initiation at completely unexpected codons was postulated. In these cases, the identified peptides encoded by the most upstream DNA match of the respective ORF was not a proteolytic product and the first codon of the most upstream DNA match was assumed to be the translation start site (Fig 9A). To find further support for the predicted sORFs, we looked for possible upstream ribosomal binding sites. Hence, eight of the newly derived sORFs are preceded by a ribosomal binding site within a distance up to 14 nucleotides upstream of the putative start codon.

Based on their genome localization, the 24 novel sORFs can be classified into three main categories: (i) located in intergenic regions, (ii) overlapping with other ORFs either with the 5′ or with the 3′-end but in a different reading frame, and (iii) located within annotated protein coding sequence but in a different reading frame. For the latter two we can additionally distinguish between those located at the same strand and those located at the complementary strand. The majority (n = 13) belongs to the group (iii) of which five are localized at the same strand (Fig 9B). This group of proteins is highly interesting and first evidence for their existence was recently reported for several other organisms using N-terminomics or ribosomal profiling in combination with retapamulin [24,28,55,56]. Seven of the newly identified sORFs were detected by at least two different approaches. Four sORFs were allocated to group (ii) and ten to group (i). Notably, three SP100 belonging to group (i) are encoded by regions within pseudogenes (Fig 10A–10C). These are sORFSaNew0004 (pseudogene NWMN_RS15675), sORFSaNew0010 (pseudogene NWMN_RS01305), and sORFSaNew0044 (pseudogene NWMN_RS04585). While for NWMN_RS01305 only one frame shift mutation leads to an

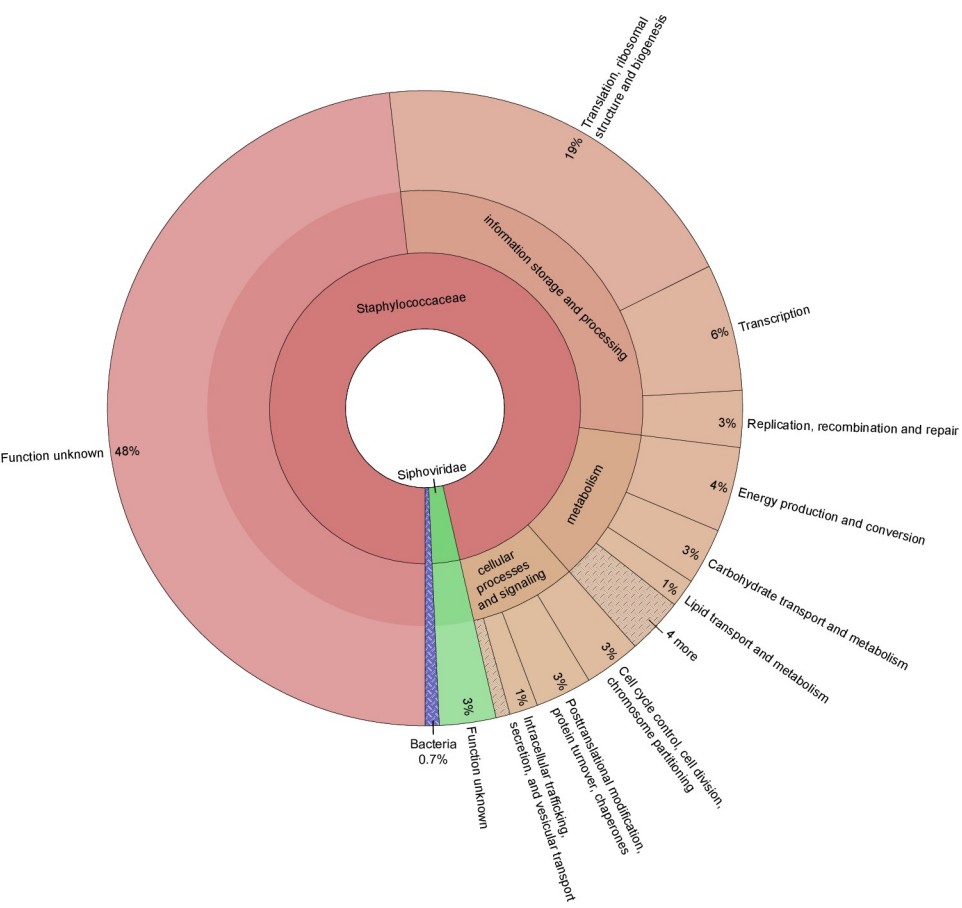

**Fig 8. Functional classification of the identified SP100.** Conserved protein domains were identified using the eggNOG database. On the basis of this, 140 SP100 (80%) were successfully assigned to an eggNOG orthologues cluster providing additional support for their existence by biological significance.

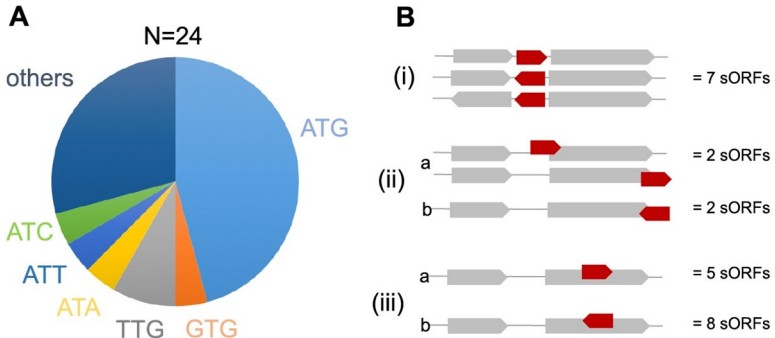

**Fig 9. Characteristics of not yet annotated SP100.** In total 24 SP100 were identified in *S. aureus* Newman, which were not covered by the used gene annotation (NC 009641.1: genome annotation from 2020-02-17). The encoding sORFs were derived by Pepper on the basis of the identified peptides and specific criteria concerning the translational start codon, the Shine Dalgarno sequence and the length of the spacer between both. (**A**) Distribution of different translational start codons between the newly predicted sORFs. (**B**) Characteristics of the genome localization of the newly predicted sORFs: (i) intergenic regions, (ii) partly overlapping with another ORF at the same strand or at the complementary strand, and (iii) completely overlapping with another ORF at the same strand or at the complementary strand.

## A    sORFSaNew0004 (25 aa)

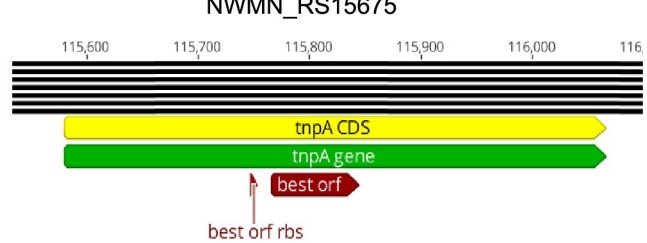

## B    sORFSaNew0010 (99 aa)

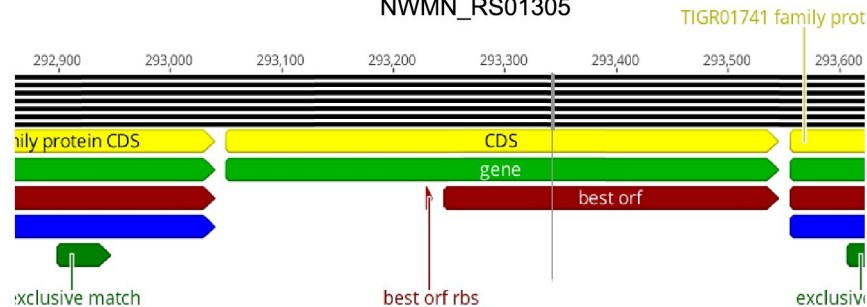

## C    sORFSaNew0044 (59 aa)

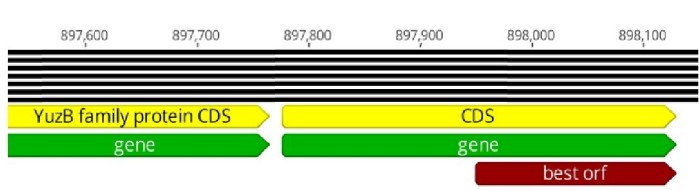

**Fig 10. Identified sORFs localized within pseudogenes.** Schematic presentation of the *NWMN_RS15675* (*tnpA)* (**A**) *NWMN_RS01305* (**B**) and *NWMN_RS04585* locus (**C**) based on the annotation of the *S. aureus Newman* genome sequence (NC_009641.1; genome annotation from 2020-02-17). Annotated pseudogenes are shown in light green and the derived coding sequence (CDS) in yellow. Matched unique peptides identified by MS/MS are depicted in dark green and the best ORF derived by Pepper on the basis of the identified unique peptides and additional features is depicted in dark red. Pepper analyses for digestion with trypsin are presented.

interruption of the open reading frame and translation of the full length protein cannot be excluded, for NWMN_RS15675 and NWMN_RS01305 several interruptions have been detected and translation of the full length protein is extremely unlikely.

## Translation of identified sORFs

We additionally performed ribosome profiling for *S. aureus* grown under the exact same growth conditions as used for assessing the proteome. Ribosome profiling provides a snapshot of translation [57] and the position of the translating ribosomes can be assessed with codon

precision [58]. The obtained sequencing reads, which represent ribosome-protected mRNA fragments (RPF), were mapped to the genome of *S. aureus* Newman and translation profiles were generated (for more details see [47]). For 135 (76%) sORFs identified by using our proteogenomics pipeline we detected RFPs. Among them are five that were missing in the used genome annotation. For an additional seven of the newly annotated sORFs, based on our approaches, reads have been mapped to the respective coding region close to the signal to noise threshold; they would require other validation experiments.

In case of sORFs embedded within protein coding regions at the same strand, we were not able to clearly assign RPFs. In *Escherichia coli*, Retapamulin-enhanced Ribo-seq analysis (Ribo-RET), which combines retapamulin specific arrests of initiating ribosomes and ribosome profiling to map translational start sites, revealed putative internal start sites in a number of genes [24,55]. However, preliminary attempts to use this technique in *S. aureus* for mapping translational initiation sites have failed so far, possibly due to an inefficient transport of retapamulin into the cell.

27 sORFs, which were identified by the proteogenomics approach, showed no translational activity at all. Among them are 15 sORFs which are localized on three different lysogenic phages. It is interesting to note, that these phages are highly similar and since we only used uniquely mapped reads (i.e. mapping to only one position of the genome), it is likely that the RPFs were discarded due to ambiguity. In sum, we have observed translational activity for the majority (76%) of the sORFs detected by our proteogenomics approach, validating the predictive power of the used MS data sets and TRDB in combination with the here developed proteogenomics tool Pepper for data analyses. (S4 Table).

## Discussion

Prediction and identification of sORFs coding for proteins smaller than 100 aa (SP100) is still very challenging from both computational and experimental points of view. In recent years, several studies started to address this issue in a systematic way by combining computational prediction and experimental validation using ribosome profiling and/or mass spectrometry [16,17,19,22,24,48,49,59–63] Proteomics based identification of sORFs relying on mass spectrometry is especially advantageous as it provides direct evidence for the existence of sORF encoded proteins and validates not only translation of sORFs but also stability of their gene products [64,65]. However, mass spectrometry based identification of sORFs is faced by several challenges: the very low number of peptides available for mass spectrometry and the dependence on amino acid reference sequences. To address this, we here developed a highly optimized and flexible data analysis pipeline for bacterial proteogenomics, covering all steps from (i) protein database generation, (ii) database search, (iii) peptide-to-genome mapping, and (iv) result visualization. The workflow is based on our bacterial proteogenomics pipeline Pepper, extended by Salt, a genome translator generating protein and peptide databases using different methods (e.g., stop-to-stop or start-to-stop translation). Pepper represents a rule based expert system that enables fully automated MS data analysis for empirical and evidence based gene annotation. Automatic rule based selection of high quality PSMs for peptide identification combined with an improved start codon detection based on N-terminal peptides distinguishes this workflow from already existing pipelines for bacterial proteogenomics [16,17,66–68]. It is thus particularly well placed to identify expressed SP100 in bacteria by one unique peptide completely independent of existing genome annotations. Compared to very sensitive ribosome profiling approaches, which are more frequently used for sORF identification, highly confident MS based identification of proteins as facilitated by this workflow

provides direct evidence for the existence of small proteins and validates not only translation of sORFs but also stability of their gene products under the tested conditions.

To evaluate commonly used experimental approaches for identification of SP100 in *S. aureus*, we applied a gel-based and a gel-free LC-MS/MS approach in combination with three different endopeptidases. Peptide identifications were obtained by MaxQuant using a six-frame "stop-to-stop" translation of the reference genome sequence. Only high-quality peptide identifications were accepted. With each approach a considerable number of unique proteins were identified. Clearly, only a combination of different approaches may facilitate the identification of the entire set of small proteins expressed in a defined bacterium [15,19,49,60]. Interestingly, using the 1D gel for protein fractionation, small proteins have been detected in almost all fractions indicating strong interactions with larger proteins. In addition, we tested the value of using different endoproteases for identification of SP100: trypsin, Lys-C and AspN. It became clear that AspN performing hydrolysis of peptide bonds at the amine site of aspartyl residues [69] was less efficient for identification of SP100, at least in *S. aureus* Newman. Notably, in *Bacillus subtilis*, Lys-C and Arg-C identified additional small proteins not identifiable with a tryptic digest [49]. Peptides generated by AspN are more frequently acidic. Worthwhile emphasizing is the fact that the pI of 26% of the identified peptides using AspN was below four as compared to only 9% of the identified peptides using Lys-C and trypsin. There is strong reason to believe that the number of basic peptides strongly impact identification of small proteins that tend to be more alkaline. For an improved identification of basic proteins it may thus be essential to consider protocols aiming at higher percentages of basic peptides.

Our combined genome-wide proteogenomics approach enabled us to identify 175 coding sOFRs of which a considerable number has not yet been described. The majority (n = 120) was detected by multiple peptides providing strong evidence for their existence (Fig 5A). The identification of another 55 SP100 relied on single high-quality PSMs. 34 out of these peptides were identified in more than one experimental approach and 21 by at least 10 MS/MS scans in at least one experimental approach (S5 Table). For 135 of the here identified sORF RPFs have been detected by ribosome profiling suggesting translational activity at the respective genomic region. This technique, however, provides only a snapshot of translational activity when using a limited number of sampling points, as is the case here, and it was thus not expected to identify translational activity for all sORFs identified by our MS-based approach. In addition, translational activity for sORFs embedded in larger ORFs at the same strand are hardly to distinguish from that of the respective larger ORFs by conventional ribosomal profiling techniques. Consequently, ribosomal profiling provided additional hints for the existence of sORFs identified by our proteogenomics workflow, however, further techniques such as antibody based techniques or spectral library-based comparisons with synthetic reference peptides have to be included in follow-up studies to validate that in particular those SP100 that have been identified here with only one unique peptide are bona fide small proteins.

Out of the 175 identified SP100, 24 were not covered by the used gene annotation. Three of them were identified by at least two unique peptides and more than 50% (n = 14) were supported by at least two experimental approaches. To evaluate the suitability of the algorithm applied by Pepper to deduce sORFs on the basis of identified peptides, we used a genome annotation of *S. aureus* Newman entirely lacking ORFs with up to 303 bp and the evidence files provided by MaxQuant for our MS based approach. In this way, 169 sORFs with up to 303 bp were deduced by Pepper of which 122 overlapped completely with sORFs annotated for *S. aureus Newman* by NCBI (NC_009641.1; genome annotation from 2020-02-17). Interestingly, for 24 sORFs, differences to sORFs annotated by NCBI have been observed with respect to their length (S6 Table). While the end of these ORFs is clearly defined by one of the

three possible stop codons, the predicted start sites vary when multiple start sites are possible and selection of one of these start sites depends on the predicting algorithm used. For selection of the best ORF, our analysis tool Pepper performs an ORF ranking based on the number of identified peptides, the nature of the start codon, the presence of a ribosomal binding site and the spacer between both (S6 Table). As aforementioned the length of the ORF is only relevant when there are two ORFs belonging to the same ORF class. Notably, 21 of these sORFs deduced by Pepper are preceded by a ribosomal binding site while only seven of the sORF variants annotated by NCBI are characterized by this feature. Three of them varied by only a few codons raising the question as to whether multiple start sites may be possible. For sORFSa-New0121 we got experimental evidence that translation starts at position 1976846 using TTG as start codon. This results in a 104 aa protein instead of the 96 aa protein annotated by NCBI (NWMN_RS10090). Additional information provided by ribosome profiling and MS based identification of N-terminal peptides are thus essential to improve ORF prediction [24,55,70–72].

The biochemical roles of most of the identified SP100 are yet to be determined as they still remained largely uncharacterized at the molecular level. Co-localization of their encoding genes with already characterized genes, presence of functional domains and/or subcellular localization of the proteins in the cell [9] may provide first hints about a possible role of these proteins in cell´s physiology and virulence. Notably, 24 of the identified SP100 are associated with the four prophage regions. Out of these 22 are encoded by φNM1, φNM2 or φNM4, which are members of the *Siphoviridae* family and highly similar. Another two are encoded by φNM3 widely distributed among the human *S. aureus* isolates. Because of high sequence similarities of φNM1, φNM2 or φNM4, 19 of the identified proteins are orthologues and were associated to eight protein families.

Some sORFs are proximal in sequence space to upstream or downstream genes encoding well characterized larger proteins with enzymatic activity such as aminopeptidase (NWMN_RS104440), glutamin amidotransferase (NWMN_RS10105), glycosyltransferase (NWMN_RS05075), ribonuclease J (NWMN_RS05355), cardiolipin synthase (NWMN_RS06950), and GTP- and ATP-binding proteins (NWMN_RS02350, NWMN_RS01520, NWMN_RS10370) involved in heme biosynthesis or DNA replication. Hence, it is reasonable to hypothesize that the physiological activity of the newly identified small proteins might be associated with the activity of those larger proteins. Moreover, we identified five SP100 similar to cold shock proteins and four proteins belonging to toxin-antitoxin systems. Most interestingly, almost half of the identified SP100 are basic implying a role in binding to more acidic cellular structures such as nucleic acids or phospholipids. Similar observations have been reported recently for small proteins identified in a simplified human gut microbiome [19]. Future work will focus on characterizing these proteins.

## Supporting information

**S1 Table. All putative ORFs are classified using different criteria (if two ORF variants share the same class, the longer ORF is preferred).**
(DOCX)

**S2 Table. Selected output information provided by Pepper.**
(PDF)

**S3 Table. Identification of SP100 using different experimental workflows.**
(XLSX)

**S4 Table. Identified SP100 in *Staphylococcus aureus* Newman.**
(XLSX)

**S5 Table. MS/MS counts of SP100 identified by one unique peptide.**
(XLSX)

**S6 Table. Annotated sORFs differently predicted by Pepper.**
(XLSX)

**S1 MS MS Spectra. High-quality MS/MS spectra of tryptic peptides unique for the identified SP100.**
(7Z)

## Acknowledgments

We thank B. Jung for technical assistance and Benjamin Heiniger (Agroscope) for help with improving S4 Table.

## Author Contributions

**Conceptualization:** Stephan Fuchs, Christian H. Ahrens, Zoya Ignatova, Susanne Engelmann.

**Data curation:** Stephan Fuchs, Martin Kucklick.

**Formal analysis:** Martin Kucklick, Erik Lehmann.

**Funding acquisition:** Stephan Fuchs, Zoya Ignatova, Susanne Engelmann.

**Investigation:** Martin Kucklick, Erik Lehmann, Alexander Beckmann, Maya Wilkens, Baban Kolte, Ayten Mustafayeva, Tobias Ludwig, Maurice Diwo.

**Methodology:** Martin Kucklick, Josef Wissing.

**Project administration:** Stephan Fuchs, Zoya Ignatova, Susanne Engelmann.

**Resources:** Lothar Jänsch, Susanne Engelmann.

**Software:** Stephan Fuchs, Alexander Beckmann.

**Supervision:** Stephan Fuchs, Zoya Ignatova, Susanne Engelmann.

**Writing – original draft:** Stephan Fuchs, Susanne Engelmann.

**Writing – review & editing:** Stephan Fuchs, Martin Kucklick, Erik Lehmann, Alexander Beckmann, Maya Wilkens, Baban Kolte, Ayten Mustafayeva, Tobias Ludwig, Maurice Diwo, Josef Wissing, Lothar Jänsch, Christian H. Ahrens, Zoya Ignatova, Susanne Engelmann.

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
