## [Decision Letter · Decision Letter 0]

28 Dec 2020

Dear Dr Engelmann,

Thank you very much for submitting your Research Article entitled 'Revealing the hidden world of small proteins in Staphylococcus aureus, a proteogenomics approach' to PLOS Genetics.

The manuscript was fully evaluated at the editorial level and by independent peer reviewers. The reviewers appreciated the attention to an important problem, but raised some substantial concerns about the current manuscript. Based on the reviews, we will not be able to accept this version of the manuscript, but we would be willing to review a revised version. We cannot, of course, promise publication at that time.

If you decide to revise the manuscript for further consideration at PLOS Genetics, please aim to resubmit within the next 60 days, unless it will take extra time to address the concerns of the reviewers, in which case we would appreciate an expected resubmission date by email to plosgenetics@plos.org.

[LINK]

Please do not hesitate to contact us if you have any concerns or questions.

Yours sincerely,

Kai Papenfort

Guest Editor

PLOS Genetics

Lotte Søgaard-Andersen

Section Editor: Prokaryotic Genetics

PLOS Genetics

Dear Dr Engelmann.

Thank you again for submitting your work to PLoS Genetics. Your manuscript has now been seen by three independent reviewers. All the reviewers found the work to be of interest, however, they all raised several concerns (see detailed comments below), which should be addressed in your revised manuscript as well as in your rebuttal letter. Given that your study might well serve as a resource for other researchers working on S. aureus, please make sure that the PRIDE data are made available (see major comment #1 of reviewer #3) and that Table S4 is updated according to the requests made by reviewer #1.

Looking forward to your revised manuscript.

Kai Papenfort

Reviewer's Responses to Questions

**Comments to the Authors:**

Reviewer #1: This is an interesting paper, which presented a strategy combining computational approaches, proteomics, and ribosome profiling to gain knowledge on the missing part of the genome, the small proteins (< 100 aa) and peptides. The authors have made a rigorous study that will be useful for the community working on Staphylococcus aureus and in more general on bacteria. Their study revealed the existence of 178 small proteins and among them, 27 were not predicted by the genome annotation of the Newman strain, with 83% having a size above 50 aa. Many of these small proteins are basic suggesting that they might target nucleic acids or phospholipids. This work opens up many perspectives for further functional studies.

I have only few comments to be addressed:

1- A significant number of the sORF encoding SP100 (almost 20%) does not present translational activity as demonstrated by the ribosome profiling data (Table S4). The sORF (below 100 nts) seems also to be poorly active as no translational activity was detected for almost 65% of them. As the authors mentioned, it is difficult to detect small proteins that are encoded inside other genes as they can be masked by the signal of the annotated gene. However, recent ribosome profiling improvements have been done to enrich ribosome density at the start codons (using for instance retapamulin to block translation initiation or Onc112 which blocks the entry at the elongation step). It would be of interest to mention these works that can improve the detection of small proteins below 50 aa.

2- The authors should better mention the limits of the proteomic approach they used. For instance, because they have used cytosolic protein fraction for proteomics, it might be possible that some of the peptides have been missed because they are either secreted or transported to the membranes.

3- The annotation of Table S4 can be significantly improved. For instance, I think that CspA protein is an RNA chaperone that is primarily involved in post-transcriptional and translational controls but not in transcriptional control. Most of the highly abundant SP100 are factors involved in ribosome function and biogenesis. Several of these genes are given as they belong to several bacterial ribosomal proteins. The authors should precise the meaningful. Does it mean that these sORFs are pseudogenes? It is also mentioned that the ORFSaNew0084 is an RNA binding protein, which binds to sRNAs and mRNAs, it is Hfq protein? The legend of Table S4 should be detailed (ORF type?).

4- Page 13: the authors have mentioned that the sORFs they have identified are classified into three categories: sORFs in intergenic regions, sORFs overlapping with the 5’ or 3’ UTR of different ORFs, and located within a larger ORF but in a different reading frames. It would be of interest to add this information in Table S4 as well as the sequences of the proteins. Did they find sORFs as possible upstream leaders of other genes? Certainly that further work help to decipher the sORFs that are translated within operons from those that specifically control the translation of the downstream gene.

5- The title has to be changed because even if the strategy is powerful, many of the small ORFs (below 50 aa) were not identified here... I would suggest something like "Towards the charactrization of the hidden world of small proteins..."

Reviewer #2: In the report, Revealing the hidden world of small proteins in Staphylococcus aureus, a

proteogenomics approach, Fuchs et al., developed a bioinformatics workflow to identify potential small protein candidates in the genome of Staphylococcus aureus. The workflow combines proteomics and genomic data analysis and is validated by ribosome profiling. The approach is already established; comparable workflows (with certain variations from lab to lab) are described to explore the existence of small proteins in different organisms (e.g. Yang 211, Miravet-verde 2019; Weaver 2019, Venturini 2020). In this regard, the workflow is solid but it does not have a strong component of innovation away from the field of Staphylococcus aureus. Indeed, for S. aureus laboratories will be an interesting tool.

The experimental part combines MS peptidic data coupled to ribosome profiling. The ribo seq experiment detected a number of ribosome-protected mRNA fragments (RPF) that were mapped to 76% of the total putative sORFs in the genome. However, only a few of them were validated with the corresponding MS peptidic signal. sORFs that are embedded within other larger ORFs did not produce a reliable RFPs signal while other sORFs showed no translational activity. Although this integrative work represents an interesting starting point for identifying small proteins in S. aureus, the strength of the experimental dataset does not allow to answer important questions as to whether these peptidic fragments detected by MS are bona fide small proteins and more importantly, whether they have any biological relevance or there is any indication that these peptides play a role in any cellular process (e.g. interaction with other proteins, phenotypes associated with their expression or virulence potential).

The authors obtained MS-based peptidic signal using gel-based and gel-free pre-fractionation methods and compared the two methods for their precision and accuracy to detect small proteins. It is unclear to this referee what is the purpose of this comparison, what are the benefits of one approach in comparison to the other and more importantly, what is the final conclusion of the comparative study. I understood that the authors used these two methods to detect common peptidic signals and used them as trustable signals. So, it seems that one must run the two approaches to obtained a reliable signal to detect small proteins.

Other comments:

1) Regarding ribosomal profiling, this technique does not corroborate the existence of the identified SP100 as this technique operates at the mRNA level. Please rewrite this sentence acordingly.

2) The number and classification of the MS data is hard to follow. I had problems to understand this part of the manuscript. For instance, there are several values of SP100 found using different approaches/sections and is at times difficult to digest.

3) Why do the authors have focused on the strain Newman? It seems that this clinical isolate is not an MRSA and does not cause important infections these days.

Reviewer #3: The authors report proteomic re-annotation of the S. aureus genome, with a combined LC-MS/MS and ribosome profiling approach leading to identification of 27 previously unannotated gene products <100 amino acids. Given the mushrooming number of recent reports of proteogenomic analysis of small proteins, including many bacteria, novelty of new studies is an important consideration. To that point, this paper most importantly offers (1) the first proteogenomic analysis of S. aureus Newman, an important pathogen, and (2) the development of a web interface for 6-frame genome translation database construction for MaxQuant searches. While many methods for 6-frame genomic databases have been previously reported, it seems worthwhile to create an accessible tool for non-specialists. In another more technical note, the application of multienzyme digest in proteogenomics has been underexplored and the demonstration that this approach (modestly) improves small protein detection in this study may be of some interest to proteomics practitioners. Other mass spectrometry methods reported, such as caparison of gel vs. non-gel-based prefractionation, are in line with previous reports using human samples (such as https://pubs.acs.org/doi/abs/10.1021/acs.analchem.6b00191).

Major concern: It is true that many proteogenomics studies rely on single peptide-spectral matches (PSMs) for small protein identification, due to the low occurrence of unique tryptic peptides within short sequences. This, in itself, is not a concern. However, it is best practice to provide the unique PSMs, along with scores, mass errors, etc., for examination in the case of single peptide identifications – for example, see http://pubsapp.acs.org/paragonplus/submission/jprobs/jprobs_proteomics_guidelines.pdf?. This is particularly important for proteins reported for the first time. I may simply have missed this – in which case, I apologize – but I was unable to access the PRIDE data or find a reviewer key, nor was I able to locate reproduced PSMs in the supplementary material. While the authors have made clear that they utilized stringent criteria, manual inspection is still required.

Minor concern:

Pages 13-14: “For the latter two we can additionally distinguish between those located

at the same strand and those located at the complementary strand. The majority (n=13) belongs to the group (iii) of which five are localized at the same strand (Fig 9). This group of proteins is highly interesting as their existence has been ignored for a long time and highlights the power of an unbiased proteogenomics approach as applied here.”

This sentence seems to imply that overlapping same-strand genes have not yet been reported, which is not true and references should be provided. There is certainly precedent for their existence – at least in E. coli and human. For example, https://www.ncbi.nlm.nih.gov/pmc/articles/PMC5761644/, https://academic.oup.com/nar/article/48/3/1029/5556081, https://academic.oup.com/nar/article/48/3/1029/5556081, https://www.embopress.org/doi/full/10.15252/embr.202050640, just to provide a few examples out of many.

**Have all data underlying the figures and results presented in the manuscript been provided?**

Reviewer #1: Yes

Reviewer #2: Yes

Reviewer #3: Yes

PLOS authors have the option to publish the peer review history of their article (what does this mean?). If published, this will include your full peer review and any attached files.

Reviewer #1: No

Reviewer #2: No

Reviewer #3: No

---

## [Decision Letter · Decision Letter 1]

5 Apr 2021

Dear Dr Engelmann,

Thank you very much for submitting your Research Article entitled 'Towards the characterization of the hidden world of small proteins in Staphylococcus aureus, a proteogenomics approach' to PLOS Genetics.

The manuscript was fully evaluated at the editorial level and by independent peer reviewers. The reviewers appreciated the attention to an important topic but identified some concerns that we ask you address in a revised manuscript

We therefore ask you to modify the manuscript according to the review recommendations. Your revisions should address the specific points made by each reviewer.

[LINK]

Yours sincerely,

Kai Papenfort

Guest Editor

PLOS Genetics

Lotte Søgaard-Andersen

Section Editor: Prokaryotic Genetics

PLOS Genetics

Dear Dr. Engelmann.

Your revised manuscript has now been evaluated and I am happy to tell you that all reviewers agree that the manuscript has significantly improved and, in principle, is suitable for publication. However, reviewer #2 identified two remaining issues that need to be addressed before we can move forward. Please make sure to address these remaining issues in your revised manuscript and your rebuttal letter. Please note that I will first assess your responses and may not need to approach the referee for further comments.

Kind regards

Kai Papenfort

Reviewer's Responses to Questions

**Comments to the Authors:**

Reviewer #1: The authors have taken into account most of the suggestions of the referees, which have improved the manuscript. The work provides a catalog of novel small proteins that will require further analysis to proove their functional importance. Nevertheless this is certainly a useful study.

Reviewer #2: This is a revised version of the manuscript; Revealing the hidden world of small proteins in Staphylococcus aureus, a proteogenomics approach by Fuchs et al. In general, I find the authors’ replies to my satisfaction. I just have two remarks related to my previous comments:

1) One of my criticism was related to the limited innovative component of the workflow proposed in this work; proteomics + genomics (salt) validated by MS (pepper). The authors replied that Pepper is the component of innovation; MS data analysis for identification of small proteins, to crosscheck this data to ORF prediction. MS analysis provides a direct detection of small proteins whereas ribosome profiling provides indirect proof. Then, I do not understand why the authors validated MS data analysis with ribosome profiling. I suggest the authors made this innovative component more clear; maybe they can extend a bit the discussion section to explain the benefits of this technique in relation to other, already existing, approaches.

2) My second point was precisely related to pepper. sORFs embedded within other larger ORFs did not produce a reliable RFPs signal whereas other sORFs showed no translational activity. I meant to emphasize that, whereas this work represents an interesting starting point to identify small proteins in S. aureus, the pepper workflow seems to not provide a sufficiently strong experimental proof to validate that all sORF detected are bona fide small proteins. I understand this is beyond the scope of this work; this referee does not mean that the authors must check and validate all the small proteins detected and whether they play a biologically relevant role. Nonetheless, I think it would be a good idea to made a clear note, maybe in the discussion section, about the limitations of the pepper workflow. I think is important to discuss not only the sORF that were validated but also why other sORF were not possible to validate. This would make the workflow more robust to the eyes of the reader.

Reviewer #3: With the clarification and inclusion of PSMs and criteria, which are indeed very high quality, I believe this manuscript well passes the bar for publication.

**Have all data underlying the figures and results presented in the manuscript been provided?**

Reviewer #1: Yes

Reviewer #2: Yes

Reviewer #3: Yes

PLOS authors have the option to publish the peer review history of their article (what does this mean?). If published, this will include your full peer review and any attached files.

Reviewer #1: No

Reviewer #2: No

Reviewer #3: No

---

## [Editor Report · Decision Letter 2]

7 May 2021

Dear Dr Engelmann,

We are pleased to inform you that your manuscript entitled "Towards the characterization of the hidden world of small proteins in Staphylococcus aureus, a proteogenomics approach" has been editorially accepted for publication in PLOS Genetics. Congratulations!

Yours sincerely,

Kai Papenfort

Guest Editor

PLOS Genetics

Lotte Søgaard-Andersen

Section Editor: Prokaryotic Genetics

PLOS Genetics

Comments from the reviewers (if applicable):

Dear Dr Engelmann.

Thank you again for submitting your revised work, which I think is now suitable for publication. Congratulations on an excellent manuscript.

Kind regards.

Kai Papenfort

(Guest Editor)

**Data Deposition**

http://datadryad.org/submit?journalID=pgenetics&manu=PGENETICS-D-20-01793R2

**Press Queries**

---

## [Editor Report · Acceptance letter]

27 May 2021

PGENETICS-D-20-01793R2 

Towards the characterization of the hidden world of small proteins in *Staphylococcus aureus*, a proteogenomics approach 

Dear Dr Engelmann, 

We are pleased to inform you that your manuscript entitled "Towards the characterization of the hidden world of small proteins in *Staphylococcus aureus*, a proteogenomics approach" has been formally accepted for publication in PLOS Genetics! Your manuscript is now with our production department and you will be notified of the publication date in due course.

With kind regards,

Kata Acsay

PLOS Genetics

On behalf of:
